# Population Demographics of Owned Dogs in Greater Bangkok and Implications for Free-Roaming Dog Population Management

**DOI:** 10.3390/ani15091263

**Published:** 2025-04-29

**Authors:** Elly Hiby, Tuntikorn Rungpatana, Alicja Izydorczyk, Valerie Benka, Craig Rooney

**Affiliations:** 1International Companion Animal Management (ICAM) Coalition, Cambridge CB23 7EJ, UK; 2Soi Dog Foundation, Phuket 83110, Thailand; tuntikorn@soidog.org (T.R.); ala@soidog.org (A.I.); 3Independent Researcher, Ann Arbor, MI 48103, USA; vbenka@gmail.com; 4Dogs Trust, London EC1V 7RQ, UK; craig.rooney@dogstrust.org.uk

**Keywords:** dog, dog keeping, owned dog, free-roaming dog, stray dog, sterilisation, rabies, dog welfare, dog abandonment, dog acquisition

## Abstract

A Knowledge, Attitudes, and Practices (KAP) survey was conducted with residents of Greater Bangkok, Thailand, to gather information about the region’s population of owned dogs, which is estimated to be close to 1.3 million. The survey explored owners’ care and keeping of their dogs, including their practices of sterilisation, rabies vaccination, veterinary care, and confinement to a home or yard. The survey also asked where owners acquired their dogs and what happened to dogs that departed the household. Respondents reported high rates of vaccination among owned dogs. Approximately half of owned dogs were reported to be sterilised, and two-thirds were allowed to roam for at least part of the day. Just under half had been adopted from the street. Findings from the study are important for planning future dog population management programmes and evaluating the impact of management efforts conducted to date on the health and welfare of owned dogs, including making sure that they are vaccinated against rabies and sterilised to prevent births of unwanted puppies. Rabies vaccination is important not only for canine health but also to prevent rabies transmission to humans.

## 1. Introduction

Worldwide, there is a multitude of canine demographics, dog-keeping practices, and, more broadly, human–canine relationships. Understanding these at local levels helps to improve dog welfare and manage dog populations by informing interventions to support owners and caretakers in providing the care necessary for the health and well-being of dogs and, by extension, the people with whom they interact.

Within the vast global population of dogs, estimated to be between 700 million [1] and 1 billion [2], owned and free-roaming populations overlap and intersect in complex and nuanced ways. In a range of locations, a large proportion of owned dogs are free to roam for at least a portion of the day or night [3,4,5,6,7,8]. These dynamics are receiving growing attention.

This study focuses on owned dogs and dog ownership in the large metropolitan region of Greater Bangkok, Thailand. To better understand the evolving numbers and dynamics of owned dogs in Greater Bangkok has intrinsic value, plus important implications for owned dog health and welfare. It also has ramifications for managing populations of free-roaming dogs and, with that, preventing cases of canine rabies that also pose a threat to human lives. These dynamics in Greater Bangkok have been understudied to date.

Greater Bangkok includes the city of Bangkok and five adjacent provinces. The region’s human population is expected to grow by approximately 35% between 2015 and 2035 and continue growing until at least 2050 [9]. Concomitant to population growth is rapid urbanisation and the creation of new residential development and commercial centres beyond the city of Bangkok and into surrounding provinces [10,11].

Alongside this growth and urbanisation of Greater Bangkok, there are indications of changing trends in dog ownership in the metropolitan region and Thailand more broadly. A recent market research study estimated that 47% of Thailand’s households own dogs, a higher percentage than in many other Asian countries [12]. The country’s pet-related businesses have also recently seen significant growth [13], and the pet care market is forecasted to continue to flourish [14,15]. Urban areas have been found to have a higher density of dogs per square kilometre compared to rural areas [16]. While high rates of dog ownership are, in part, attributed to rapid growth in affluent urban areas [17], half of the households in a rural Northern Thailand survey also reported owning dogs [18].

Limited research studies over the past two decades have explored Thailand’s owned dogs [16,18,19,20,21], and fewer still [16] have focused on owned dogs in Greater Bangkok. Most of these studies have found that over half of owned dogs in Thailand roam freely for part of, if not all, the time [16,18,19]. Studies have reported varied but overall low rates of sterilisation among owned dogs [18,21] and particularly among males [22]. Rabies vaccination rates in owned dogs are higher than sterilisation [18,19,21]. However, vaccination rates and vulnerability to rabies also vary among subpopulations of owned dogs, with studies finding that roaming owned dogs had lower vaccination rates than their confined counterparts [19] and that most dogs reported to have been infected with rabies were unvaccinated owned dogs that roamed some or all of the time [16].

Dog ownership in Thailand has long been linked to dog abandonment [22,23,24]. Data from street surveys conducted in Greater Bangkok suggest that free-roaming dog numbers are sustained, in part, by owned dog populations through roaming, abandonment, or loss [8]. This is consistent with a recent spatial analysis in Thailand, which found that areas with more owned dogs, such as Bangkok, also had more ownerless dogs [16].

The complex dynamics and overlap between subpopulations of dogs, specifically owned and free-roaming, have implications for efforts to manage dog populations and ensure the high levels of rabies immunity required to prevent rabies spread among dogs that roam freely. Between 2016 and 2023, the Thai animal welfare organisation Soi Dog Foundation, in partnership with the UK-based animal welfare organisation Dogs Trust, sterilised and vaccinated over 400,000 dogs in Greater Bangkok [8]. This was accomplished through an intensive rotational Catch, Neuter, Vaccinate, Return (CNVR) programme aimed at managing the free-roaming dog population through non-lethal means [8]. CNVR programs, sometimes referred to as Animal Birth Control (ABC), have been implemented elsewhere in the world in areas with free-roaming dogs [25,26,27], and their success at managing and stabilising free-roaming dog numbers and preventing rabies in dogs and humans is supported by data. The presence of large numbers of owned roaming dogs in Greater Bangkok required the CNVR intervention to establish protocols to include owned dogs (with owner consent) in their efforts, in addition to the usual CNVR targets of unowned and community-owned animals. However, optimising free-roaming dog population management and rabies prevention will also require attending to owned dogs through additional measures. Doing this effectively necessitates understanding more about both the dogs and their owners.

This study reports the findings from a Knowledge, Attitudes, and Practices (KAP) survey of Greater Bangkok residents conducted in July and August of 2023. It explores the numbers and demographics of the dog population owned by this sample of residents. It also looks at owners’ attitudes and reported behaviours related to their dogs: the acquisition and loss of owned dogs, plus caretaking by owners in terms of confinement practices, sterilisation, rabies vaccination, and other veterinary care. The study also considers associations between dog owners’ behaviours and practices and the extent of CNVR implementation in the areas where these owners live. A companion paper [28] considers residents’ experience with and behaviours towards free-roaming dogs in relation to the CNVR intervention.

## 2. Materials and Methods

### 2.1. Study Area

Greater Bangkok, also known as the Bangkok Metropolitan Region, includes the city and province of Bangkok plus five adjacent provinces: Nakhon Pathom, Pathum Thani, Nonthaburi, Samut Prakan, and Samut Sakhon. There are 50 “khets” (districts) in Bangkok, and 309 subdistricts in the other 5 provinces, totalling 359 administrative areas of similar land area. These 6 provinces cover nearly 8000 km^2^ [11], and in 2022, when this KAP study was being designed, they were home to 11 million people, with provincial populations ranging from approximately 589,000 (Samut Sakhon) to 5.5 million people (Bangkok) [29]. Bangkok has both the largest human population and the highest population density. The percentage of Thailand’s population living in urban areas has grown rapidly in the past two decades [30], which is reflected in Greater Bangkok’s concomitant population growth [31].

### 2.2. Survey Design

The survey consisted of 42 questions, 20 of which focused on owned dogs. The remaining questions focused on perceptions and experiences with free-roaming dogs and respondent demographics [28]. Most questions were multiple choice, sometimes with an option to provide further details following a response of “other”. For each interview, the interviewer also recorded the GPS location (latitude, longitude, altitude, and accuracy) and selected the khet or subdistrict name from a list of 100 options. Interviewers did not proactively state their Soi Dog Foundation affiliation but answered truthfully if asked. They then noted this disclosure.

The 20 questions about owned dogs included asking respondents the following: Do they own any dogs? If so, how many, and how were they acquired? How are their dogs kept (confinement practices)? Are their dogs sterilised and, if not, why not? Were their dogs vaccinated against rabies in the past 12 months and, if not, why not? Have they taken them to the veterinarian in the past year and, if so, for what reason? Respondents were also asked if any dogs left their household in the past year; if they answered affirmatively, they were asked what happened to the dog? Dogs that had been vaccinated against rabies within the prior 12 months were categorized as “vaccinated.” “Unvaccinated” dogs included both those who had never been vaccinated and those who had not been vaccinated within the past year. The term “soi” used in the questions refers to the Thai word for local residential streets and equates to a respondent’s local neighbourhood. The questionnaire is available in Appendix A.

### 2.3. Informed Consent and Confidentiality

Before beginning the survey, interviewers introduced themselves by their first names. They explained that the purpose of the interview was to better understand dogs in their community, there were no essential questions, respondents could skip any question, and there was neither a cost nor compensation for participating. Interviewers also explained that the data would be shared with organisations that can provide services for dogs and with consultants who would analyse the data.

Respondents were required to be at least 18 years of age to take part in the survey independently; if under 18, they could participate if supervised by an adult from within their household. Due to the content of the questions, respondents were also required to have lived in the household for at least one year.

The survey was completed without asking respondents for their names. At the end of the survey, the respondent was asked if they would like to share their name and contact information for follow-up or remain anonymous.

### 2.4. Data Collection

The survey was conducted in Thai using face-to-face interviews between 21 July and 27 August 2023. The 20 interviewers worked in pairs, entering the responses given during the interview into a smartphone App for data collection [32]. Stratified random sampling was used to select respondents. Administrative areas were categorised according to how many rounds of CNVR had been performed in that area between 2016 and 2023. There were either one, two, or three rounds conducted in each area with CNVR programming; areas with zero rounds served as controls. Twenty-five random areas were selected from each of these categories, creating a sample of 100 administrative areas for the survey. Within each sample area, 16 random GPS points were created from areas identified to be residential using High-Resolution Population Density Maps and Demographic Estimates [33]. This showed human population density as “raster cubes”, in which a density greater than one was assumed to be a residential building. A 30 m buffer was added to each cube. Sixteen random points were created along the roads that fell within these buffered populated raster cubes. A Google map link was created for each random point to enable an interview team of two people to travel directly to this point. The interview team then selected the two households closest to each of these points for interviews. If no one was at home or the resident refused to take part, the next closest household was selected until two surveys were completed for the point. This returned a sample of approximately 800 household respondents per category of CNVR rounds. This sample size was calculated on the assumption that 20% of households would own an average of 1.5 dogs, resulting in 240 owned dogs from which the target sterilisation and vaccination coverage of 80% could be measured with 5% error and 95% confidence, a core metric of interest related to owned dogs.

### 2.5. Data Analysis

The sex and age of the owned dog, roaming versus confinement status, province, and rounds of CNVR within the respondent’s subdistrict (or “khet” in Bangkok) were tested for their ability to predict various binomial outcomes. Outcomes of interest included whether a dog was confined or allowed to roam, sterilised or intact (unsterilised), vaccinated against rabies in the past year or unvaccinated, and a recipient of veterinary care within the past year or not attended to by a vet. Descriptive statistics are given for the sources of the dogs and the fate of dogs no longer in the household.

Data were analysed using binomial Generalised Linear Mixed Models (GLMMs). This enabled testing for the fixed effect of potential predictors, including the sex and age of dogs and number of CNVR rounds, on outcomes while including the khet or subdistrict in the model as a random effect. This addressed the possibility that respondents living in the same khet or subdistrict may share similar characteristics related to their location and improved the power of the analysis.

A province could also be included as an additional random effect along with a khet or subdistrict. However, for most outcomes of interest, the addition of the province led to very small increases in the model fit, as measured by Akaike Information Criterion (AIC) values, and for some outcomes, the fit was not as good as with the khet or subdistrict alone.

The data were uploaded to KoboToolbox during collection, then downloaded to Excel and analysed using R version 4.3.2 via RStudio [34,35] and package lme4.

### 2.6. Ethical Approval

All procedures were in accordance with the 1964 Helsinki Declaration and its later amendments or comparable ethical standards. In addition, all methods were approved by the Dogs Trust Ethical Review Board of the Dogs Trust UK (reference number ERB037 on 21 October 2020).

## 3. Results

### 3.1. Respondents

A total of 3437 households were approached, and someone was at home in 3304 of them. Of this latter group, 99 households declined to take part in the survey immediately, and two declined after hearing the consent statement, generating a 93% response rate. Among the 3203 consenting respondents, 303 (9.5%) asked for the interviewer’s affiliation and were provided this information.

Respondents were evenly divided between females (50.1%) and males (49.8%). Their average age was 46 years (range: 13–85). Sixteen respondents (0.5%) were under 18 years old and had adult supervision during the interview.

### 3.2. Dog Ownership

In this sample, 628 out of 3203 households (19.6%) owned a total of 1138 dogs (Figure 1). This included one household with 80 dogs that was excluded from the following statistics because it was an extreme outlier (the Soi Dog Foundation also followed up to provide veterinary care to the dogs at this address). The mean number of dogs per dog-owning household was 1.68, with a standard deviation of 1.49 (range: 1–14) (Table 1).

Dog ownership varied by province, with significantly higher percentages of dog owners in Nakhon Pathom (30.5%) and Pathum Thani (29.8%) than in the other provinces. These two districts also had the highest numbers of owned dogs per dog-owning household, as well as the highest dog density (human-to-dog ratio of 4:1 and 3:1, respectively, versus a minimum of 10:1 in all other provinces). In Bangkok, the most populous and densely populated province studied, approximately 1 in 10 survey respondents (9.4%) reported owning dogs. Extrapolating these data translates to an estimated range of approximately 59,000 to 444,000 owned dogs in each province, and a total of nearly 1.3 million owned dogs in Greater Bangkok (Table 1).

### 3.3. Owned Dog Demographics

Demographic information was provided by owners for 69% (783) of the 1138 owned dogs in the survey. If the owner did not know the dog’s age or sex, a category of “unknown” was recorded. Among these 783 dogs, 63% (494) were reported to be male, and 37% (289) were reported female, constituting a male-to-female ratio of 1.7:1. Among the 723 dogs for which sex and age were reported, the mean age was 4.2 years (SD = 3.0 years), with a maximum of 17 years; 8.4% of dogs were younger than 1 year old (Figure 2).

### 3.4. Owned Dog Keeping Practices

A comprehensive visualisation of confinement, sterilisation, and vaccination practices for owned male and female dogs in this sample is presented in Figure 3. The details of care and keeping practices, plus dog acquisition and loss from households, are reported below.

### 3.5. Confinement and Roaming Behaviours

Among the 785 dogs in the sample for which confinement and roaming information was provided, 34% (268) were always confined. The remaining 66% (517) were allowed to roam. Approximately three-quarters of roaming dogs did so at all times, and about one-fifth roamed only during the day (Figure 4).

Confinement and roaming practices were not related to a dog’s sex. Increasing age was associated with lower odds of roaming, although the effect size per year of increasing age was small (OR 0.94, 95% CI 0.89–1.00, *p* < 0.05; Table 2).

Any level of CNVR effort in a khet or subdistrict was associated with lower odds of roaming versus dogs living in control areas. In khets or subdistricts that had experienced one or two rounds of CNVR, the odds of roaming were around half those of dogs in control areas. This was a statistical trend at *p* < 0.1. In areas with three rounds of CNVR, the odds of roaming were over 70% lower than for dogs in control areas. This was a statistically significant difference (Table 2).

### 3.6. Sterilisation

Respondents reported the sterilisation status of 783 dogs (289 females and 494 males). Among these dogs, 53% (409) were sterilised. A dog’s sex was found to be a significant predictor of sterilisation. A total of 65% of females were spayed, and 44% of males were castrated. The odds ratio for females being sterilised relative to males was 3.25 (95% CI 1.24–4.71; *p* < 0.001). Confinement status was also found to be a significant predictor. Roaming dogs had 50% higher odds of being sterilised than their confined counterparts (OR 1.50, 95% CI 1.02–2.19; *p* < 0.05). Dogs living in a khet or subdistrict with any CNVR had higher rates of sterilisation than dogs in control areas (Figure 5). An owned dog in an area with one round of CNVR had more than three times the odds of being sterilised than an owned dog living in a control area, and the odds increased for owned dogs living in areas with more rounds of CNVR (Table 3).

Owners of sterilised dogs were asked where they had their dog sterilised, and 408 answered this question. Clinics run by non-governmental organisations (NGOs) were used for 67% of dogs, followed by private (25%) and government clinics (6%).

Owners of intact dogs were asked to provide the primary reason that their dog had not been sterilised, with 365 responses recorded. For male dogs, not needing to sterilise male dogs was overwhelmingly cited as the reason for not sterilising (44%), followed distantly by not knowing about sterilisation (10%) and no perceived need due to the dog being confined (9%). For female dogs, the use of contraception was the most common reason for not sterilising (20%), followed by not knowing about sterilisation (18%) and the lack of a skilled veterinarian to perform the surgery (11%). For dogs of both sexes, less than 10% of owners attributed their decision to wanting puppies, wanting to breed their dog, perceiving surgery to be cruel, fearing unwanted behaviour changes, or expense.

### 3.7. Rabies Vaccination

Owners shared the rabies vaccination status of 776 dogs, of which 84% (658) had been vaccinated against rabies in the past year and 15% (118) had not. The vaccination status was unknown for nine dogs. Vaccination coverage ranged from a low of 79% in Samut Prakan to a high of 94% in Nonthaburi. Bangkok’s vaccination rates were also high, with 92% of dogs reported to have been vaccinated against rabies in the past year.

A dog’s age was a significant predictor of current rabies vaccination, with each year of age increasing the odds of being vaccinated (OR 1.21, 95% CI 1.10–1.33; *p* < 0.001). Dogs under 1 year old had particularly low vaccination coverage compared to their older counterparts (Figure 6).

Confinement was also found to be a significant predictor of vaccination status. Owned dogs allowed to roam had odds of being vaccinated that were about half that of dogs that were confined (OR 0.55, CI 0.32–0.95; *p* < 0.05). The rate of vaccination in confined dogs was 89%, and in roaming dogs, it was 83%.

Within this sample, a dog’s sex was not found to predict vaccination status. CNVR levels were similarly not found to predict vaccination coverage.

For dogs that had not been vaccinated against rabies within the last year, owners were asked to cite the primary reason why not, and 118 responses were recorded. Lack of transport and inability to handle the dog were each cited by 22% of those who answered this question (26 people each). Not knowing where to access vaccination was selected by 19% (22 people); not knowing about vaccination by 13% (15 people); and the dog being too young by 8% (10 people). Other answer options (too expensive, not necessary, too far to access vaccination, and considering the vaccine dangerous) were each selected by five or fewer respondents.

Of the 22 dogs whose owners reported they did not know where to access vaccination, the majority (13) lived in Pathum Thani, which also had the highest overall rate of dog ownership in this sample. All dogs whose owners reported them to be too young were between 1 month and 2 years old, with a median age of 4 months.

### 3.8. Veterinary Care of Owned Dogs

Owners were asked if their dog had been taken to a veterinary clinic within the past year. Responses were obtained for 775 dogs in the sample, of which 16% (119) had seen a veterinarian and 84% (642) had not. The owners of 14 additional dogs did not know.

Confinement status was found to be a significant predictor of visiting a veterinary clinic. Owned roaming dogs had much lower odds of receiving this veterinary care than owned confined dogs (OR 0.26, 95% CI 0.15–0.44; *p* < 0.001). Among confined dogs, 27% had received veterinary care within the last year, and 73% had not. Among owned roaming dogs, 10% had received veterinary care. The dog’s sex, age, or the rounds of CNVR within their khet/subdistrict were not predictive of veterinary clinic use.

The reasons that owners sought veterinary care for their dogs also varied by confinement practices. Although preventative care was the most common reason for a veterinary visit for all dogs, roaming dogs were taken to treat disease or injury and to be sterilised more often than their confined counterparts.

### 3.9. Owned Dog Acquisition

Survey respondents were asked where they acquired their dogs. Adoption from the street was most common, reported for 46% (347) of owned dogs in the sample, followed by receipt as a gift, reported for 37% (277 dogs). All other acquisition sources were much less prevalent: 7% percent of dogs (53) were offspring of another dog belonging to the owner, 7% (55) were purchased from a breeder, 3% (24) came from a pet shop, and less than 1% (2) were adopted from a shelter.

Owners were asked if they had acquired their dog within the last year. This served as an indicator for measuring change in acquisition over time. Approximately one-third (34%, 256 dogs) were acquired in this timeframe, and 61% (464) had been acquired more than one year prior. Five percent of respondents (37 people) did not share the timing. Sources of dogs in this sample vary by when they were acquired. A lower proportion of dogs were adopted from the street within the last year compared to more than one year prior (38% versus 50%), and more were received as gifts within the past year (44% versus 32%). The other, less common source options were less impacted by the timing of the acquisition.

### 3.10. Owned Dog Departure from the Household

All survey respondents (i.e., not only the current dog owners) were asked if any dogs had left their household within the last year, and 5% (168) responded affirmatively. For most of these households, one dog had left, but the number was as high as 10, and a total of 230 dogs were reported to have left their households within the past year. This is 26 fewer dogs than were acquired within the past year in this sample.

Owners provided additional information for 190 of these 230 dogs. The majority, 52% (100), were male, 27% (51) were female, and sex was unknown for 20% (39). Owners were asked about age at departure: 64% of dogs (122) were adults aged one or older, 10% (18) were juveniles 6–12 months old, and 12% (22) were puppies. Age was unknown for 15% (28). Puppies leaving the household were most often female, and adults leaving were most often male. Owners attributed nearly all (95%, 182 dogs) departures to death (86%) or disappearance (9%). Just 3% (five dogs) were given away. Among reported deaths, the largest proportion (41%, 79 dogs) was ascribed to road accidents, followed by illness (26%, 50 dogs) (Figure 7).

## 4. Discussion

This study explores dog ownership numbers, plus the care and keeping practices of owned dogs in Greater Bangkok. It helps to address the gap in information on owned dog populations in this large metropolitan area, which is home to over 11 million people and an estimated population of nearly 1.3 million dogs. A better understanding of owned dogs is important, given the changing practices and trends in dog ownership in Thailand [12,13,17]. It is also important given the significant number of owned dogs that roam freely here and around the globe [8,16,18,21]. In addition, this study reports frequent movement between owned and unowned subpopulations through adoption from the street and abandonment or loss. Hence, owned dogs must be considered in plans to manage free-roaming dog populations, and those that are not always confined are particularly important to achieving effective vaccination coverage to prevent the spread of canine rabies through the free-roaming dog population. In turn, such planning requires more knowledge of both owned dogs and their owners or caretakers.

The present survey was designed to gather data on Greater Bangkok residents’ attitudes and reported behaviours regarding both the dogs that they own and the street dogs in their community. This manuscript focuses on survey findings related to owned dogs. A companion paper [28] focuses on street dogs.

This study is consistent with others in its finding that the majority (66%) of owned dogs roam, and the majority (75%) of these roaming dogs do so at all times. Much smaller portions roam only during the day or only at night. The relatively flat age structure of dogs in this sample suggests that their reproduction is under reasonable control and their lifespan is reasonably long. Respondents reported the ages of 69% of the owned dogs. If a respondent did not know the age, it was recorded as “unknown”. We expect that respondents who had adopted their dog from the streets as a puppy would know the animal’s age within a range of one to two months. However, dogs adopted from the streets as adults are unlikely to be of known age and so would have been excluded from this analysis of age structure. In populations without fertility control, the largest group tends to be under one year old. This is not evident here, where only 8.4% of dogs in the sample were younger than one year old, and the mean age was 4.2 years. This indicates a more managed and stable canine population than elsewhere in Thailand, where the proportion of owned dogs under one year old approached 30%, and the sterilisation rates were 12.4% [21].

The sex of dogs in this sample skews male, with approximately 1.7 males reported for every female. This is consistent with studies conducted elsewhere in Thailand over the past two decades, including in areas with lower rates of sterilisation [19,21]. A preference for male dogs has also been observed in several other cities and regions with large numbers of owned roaming dogs [4,5,26,36,37,38,39]. This is likely due to a combination of factors, including the roles that dogs serve in households, perceived temperaments of the two sexes, and concerns about pregnancy where sterilisation is not widely accessible, affordable, or embraced. It is also consistent with the findings in this study that puppies leaving households were most often female, female dogs had over three times the odds of being sterilised than males, and among intact females, the most common reason for not sterilising was the use of contraception as an alternative method of controlling reproduction.

Among dogs whose reproductive status was shared, more than half were sterilised. In addition to skewing female, dogs allowed to roam also had 50% higher odds of being sterilised compared to their confined counterparts. Over two-thirds of sterilised dogs were reported to have had the surgery in an NGO-operated clinic. This relatively high rate of sterilisation is likely due, in part, to the large number of sterilisations carried out as part of the CNVR project led by the Soi Dog Foundation. By September 2021, they had sterilised nearly 50,000 dogs reported to be owned and roaming, 25% of their total CNVR surgeries [8].

Nearly 85% of all dogs in this survey were reported to be vaccinated against rabies in the past year. The high vaccination coverage in this sample of owned dogs is reassuring. While it is possible that these numbers are inflated (e.g., an owner may not admit that their dog is not vaccinated, or if their dog was vaccinated, it was more than a year prior), this is still a high level. It is well above the vaccination levels reported in other areas of Thailand [8] but similar to those achieved in Bali following a similar intervention programme [40].

Of concern, however, is the fact that dogs under one year old had a low rate of vaccination (61%) compared to adults, a trend identified in other countries as well [41]. Also of concern is that owned roaming dogs had lower rates of vaccination than dogs that were confined. This is counter to epidemiological relevance for rabies transmission, as free-roaming dogs have far higher contact rates than confined dogs. It is, however, consistent with other studies in Thailand [16,19] and elsewhere in the world [41]. Roaming owned dogs also had much lower odds of receiving veterinary care than confined dogs, which has health and welfare implications. These findings warrant particular attention.

The sourcing and acquisition of dogs in this sample are notable. Among the owners who answered, nearly half had adopted their dogs from the streets, and 37% had received their dogs as a gift. Respondents were not asked where the dogs they were gifted had been acquired, but it is likely that some of these gifted dogs were also originally from the streets, while others were sought from a shelter, breeder, or pet store, and others were offspring born to an existing owned dog and given away. That only two dogs were reported to have been adopted from a shelter appears remarkably low, considering the obvious popularity of adoption from the streets and that several shelters currently function within Greater Bangkok.

The sources of dogs also varied somewhat according to when they were acquired. There was a lower proportion of dogs adopted from the streets within the last year, and slightly more dogs were received as gifts. This could be due to fewer dogs on the streets in recent years [8].

The survey findings regarding owned dogs that left the household were striking. In this sample, 5% of owners reported that an owned dog (or dogs, as many as 10 in one instance) had left their household in the past year. This statistic is similar to another recent study in which 4% of respondents living in Thailand had a dog go missing in the past 12 months [20]. Among the departed dogs in Greater Bangkok, of which the sex was shared, twice as many were male as female. Puppies leaving the household were most often female, and adults leaving were most often male, and death or disappearance was the overwhelming cause.

Given that the survey was multiple choice, it was not possible to explore the reasons for, or details about, departing dogs more fully. Some respondents also may not have been entirely forthcoming about some behaviours or actions leading to dogs departing, particularly if they knew that the interviewers were from an animal welfare organisation. As such, it can probably be assumed that more dogs left households than were acknowledged. It can also probably be assumed that abandonment happens more frequently than was captured in this survey, particularly given the grey area between dogs who are abandoned and those roaming at all times without care from their owner. Similarly, the survey did not explore details around the acquisition of dogs from the streets and what the relationship might be to dogs who are lost or have disappeared. This, and the questions around the potential abandonment of dogs, would benefit from further study.

The findings in this survey also point to differences between provinces. The percentage of dog-owning households in each province ranged from approximately 9% to 30%, with Bangkok having among the lowest rates of dog ownership but the highest estimated number of overall dogs due to its human population density. Bangkok was also one of only two provinces, the other being Samut Sakhon, with a nearly even split between dogs kept confined and dogs allowed to roam. Variations in how dogs are kept and cared for reflect a need for flexibility when developing interventions for owned dogs, both roaming and confined. Variations in dog-keeping practices in Bangkok have also previously been attributed to culture and socioeconomics [23], and exploration of these factors was beyond the scope of this study.

The care and keeping of dogs in Greater Bangkok appeared to be influenced by CNVR in meaningful ways. Owners living in areas with CNVR were less likely to allow their dogs to roam and more likely to have sterilised dogs. As the design of the study was cross-sectional as opposed to longitudinal, and CNVR was launched in Bangkok and had been implemented to differing extents across Greater Bangkok at the time of the KAP survey, we cannot rule out that there were pre-existing differences in owner behaviours. However, there was a dose–response relationship. As the number of CNVR rounds rose, so did the odds of sterilising and confining one’s dog. Higher sterilisation levels make sense, given that CNVR mobile clinics treat owned dogs with the owners’ consent. Rabies vaccination levels were high for owned dogs overall, and there was no significant increase in CNVR areas. This is presumably because the local government and private veterinarians offer vaccination, so owners were not relying on the CNVR programme. The relationships between CNVR and the care of owned dogs warrant further exploration.

One possible limitation of this study is that respondents might have given responses they thought the interviewers wanted to hear. The study design tried to reduce social desirability bias by not including Soi Dog Foundation in the consent statement, but in slightly less than 10% of cases, the respondent asked for and learned the interviewer’s affiliation.

The results of this survey are particularly interesting given the changes underway in Greater Bangkok. This survey found lower rates of dog ownership among Greater Bangkok households than a study of dog ownership in rural Northern Thailand [18] or market research across the country [12]. This is likely due, at least in part, to different sampling methods used in each study. The growth in Thailand’s pet-related businesses [13] and pet care market forecasts [14,15] suggests a major uptick in pet keeping, largely correlated with the growth of, and increasing wealth in, urban areas [17]. Thailand’s urban areas have been found to have a higher density of dogs per square kilometre compared to rural areas [16], and as Greater Bangkok adds more human residents, its numbers and density of dogs will rise, as well.

It has been noted that the growth in dog ownership has also led to issues with abandonment and ownerless dogs [16,22]. The roles that dogs play in people’s lives and the impact that these have on how dogs are acquired, kept, and cared for continue to unfold. Further research could help to tease out this question. It would be valuable to explore the breeds of dogs that people own, and the purposes that dogs serve in people’s lives, and how these correlate with confinement practices and care (sterilisation, rabies vaccination, and other veterinary visits) that the dogs receive. There would be value in better understanding owners’ decision-making processes regarding confinement versus allowing a dog to roam, particularly how the choice relates to the dogs’ age and sterilisation status, and what the underlying motivators are for or against either decision.

There would also be value in further exploring the dogs adopted from the streets. Interviews with open-ended questions could elucidate owners’ motivations for adopting from the streets and the implications this has on the circulation of dogs within the community. Given the large number of dogs received as gifts, it would also be useful to better understand the origins of dogs acquired in this way. In a rural area of Northern Thailand, over 70% of survey respondents had acquired at least one dog from their neighbours [18]. In this Greater Bangkok study, a mere 5% of those who had a dog leave their household in the past year had given a dog away. This might suggest that gifted dogs are coming from sources other than those born in neighbouring homes, but more data are needed to begin to understand these potentially complex dynamics.

## 5. Conclusions

This study generates important data on owners’ behaviours and care of owned dogs in Greater Bangkok, both those that are confined and those allowed to roam. The findings of this survey align with prior studies in that a high percentage of free-roaming dogs have owners and, thus, support the need to focus on owners and their owned dogs to manage free-roaming populations. The findings suggest that dog population management efforts using CNVR have increased sterilisation rates and decreased the roaming of owned dogs, whilst vaccination levels remain high throughout Greater Bangkok, regardless of the level of CNVR effort. The study found some notable differences in the preventive care that confined versus roaming dogs receive. A dog’s sex, age, and location within Greater Bangkok were also correlated with some differences in care and keeping practices. There are several areas of further research that could expand on this study’s findings in important ways, as well as help to better understand the ways in which people acquire dogs in Greater Bangkok, and when, how, and why dogs are given away or abandoned.

## Figures and Tables

**Figure 1 animals-15-01263-f001:**
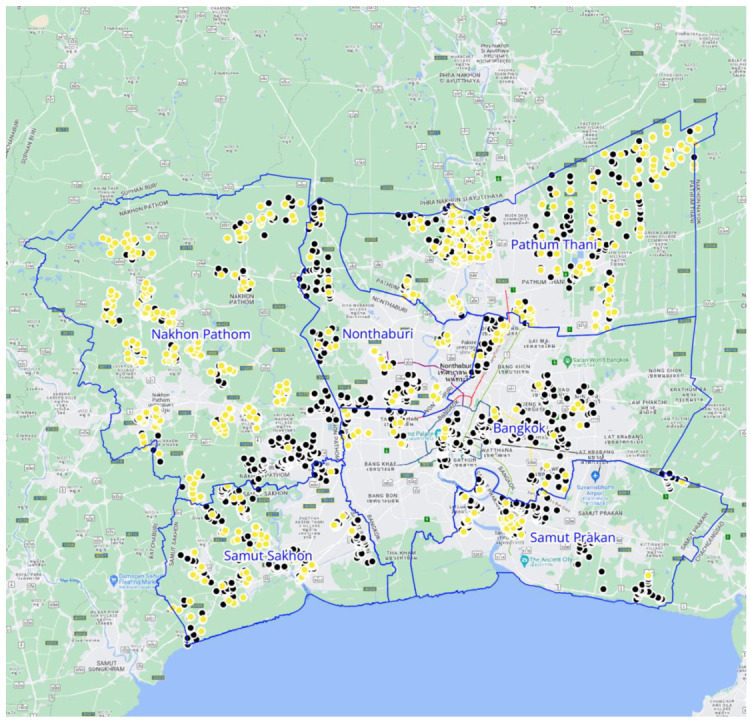
Location of households surveyed. Yellow dots indicate dog-owning households; black dots are non-dog owners. Blue lines denote boundaries of the six provinces.

**Figure 2 animals-15-01263-f002:**
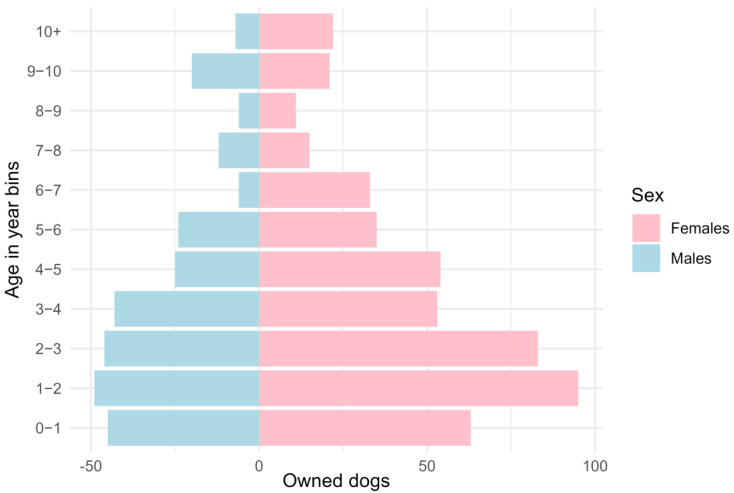
Age pyramid for 485 males and 238 females.

**Figure 3 animals-15-01263-f003:**
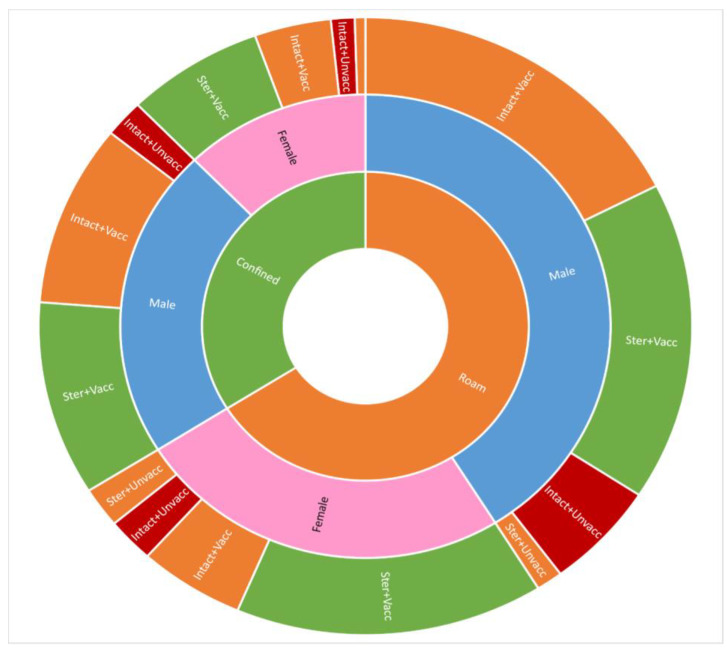
Owned dog demography and primary care practices. “Ster” = sterilised, “Intact” = unsterilised, “Vacc” = vaccinated, “Unvacc” = unvaccinated.

**Figure 4 animals-15-01263-f004:**
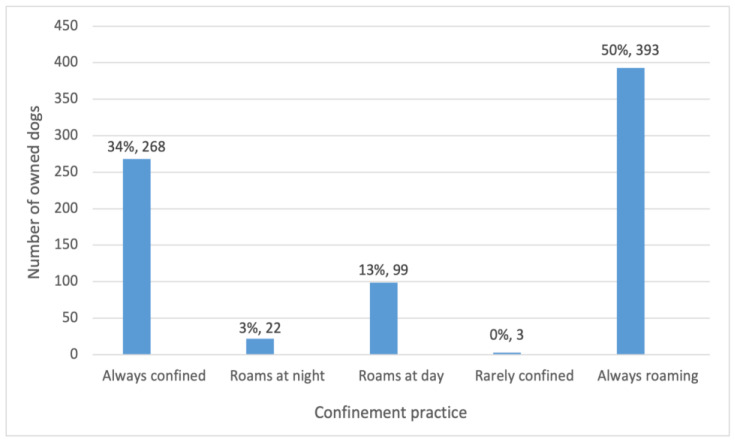
Confinement practices for 785 owned dogs, from always confined to always roaming.

**Figure 5 animals-15-01263-f005:**
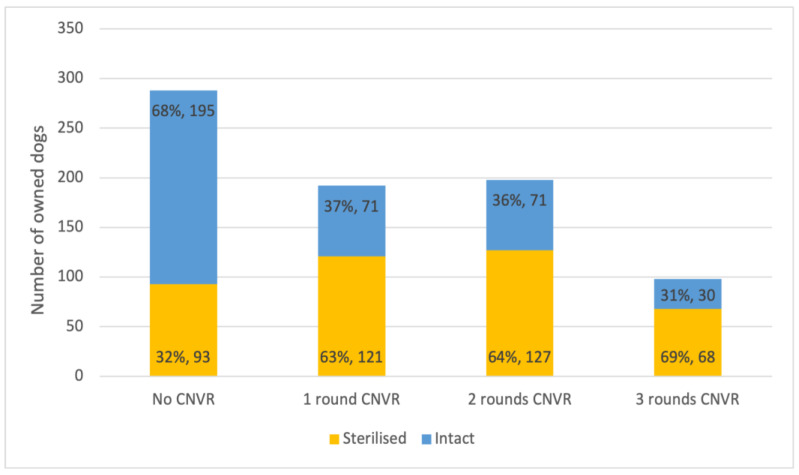
Percentage (and number in data labels) of sterilised versus intact dogs owned by households located in khets or subdistricts that have received 0, 1, 2, or 3 rounds of CNVR.

**Figure 6 animals-15-01263-f006:**
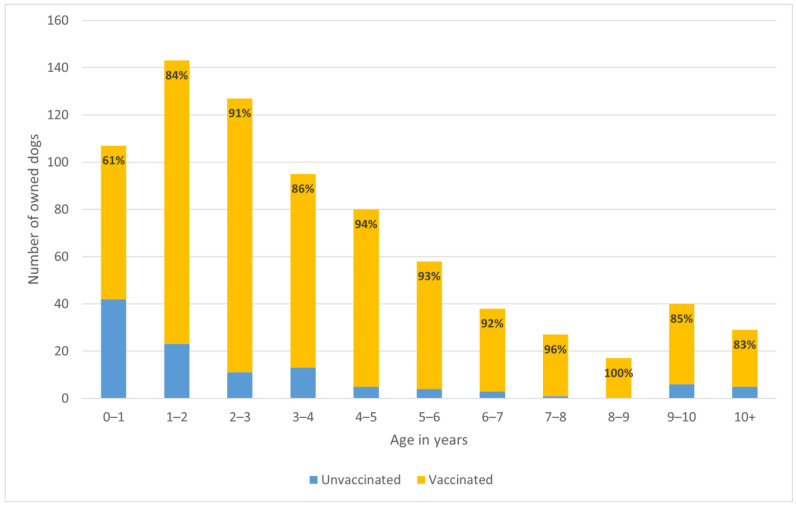
Vaccination status for each age category for 761 dogs whose owners reported both their dogs’ vaccination status and age. The data label shows the percentage of dogs in each age group vaccinated against rabies within the past year.

**Figure 7 animals-15-01263-f007:**
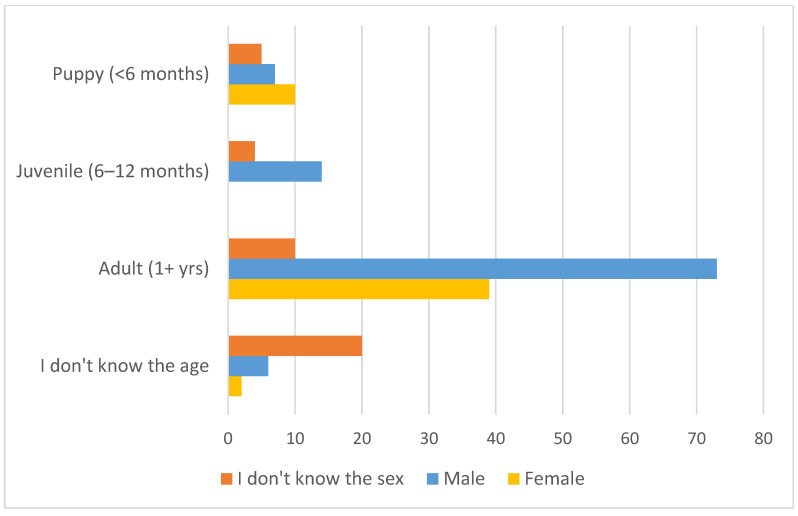
Age and sex of 190 dogs reported to have left their households within the past year.

**Table 1 animals-15-01263-t001:** Estimates of human populations and dog ownership and husbandry for the six provinces in Greater Bangkok and in total.

	Bangkok	Nakhon Pathom	Nonthaburi	Pathum Thani	Samut Prakan	Samut Sakhon	Total
Dog-owning households (%)	9.4%	30.5%	7.8%	29.8%	9.9%	13.8%	13.2%
Non-owner households (%)	90.6%	69.5%	92.2%	70.2%	90.1%	86.2%	86.8%
Owned dogs per dog-owning household (mean)	1.47	1.88	1.17	1.89	1.28	1.38	1.6
Number of households	3,197,865	426,205	739,849	677,046	750,422	309,823	6,101,210
Human population	5,494,932	921,882	1,295,916	1,201,532	1,360,227	589,428	10,863,917
Owned dogs that roam (%)	46.9%	66.4%	63.8%	73.3%	36.4%	50.5%	58.7%
Owned dogs always confined	53.1%	33.6%	36.2%	26.7%	63.6%	49.5%	41.3%
Sterilised	65.3%	67.0%	67.1%	50.6%	59.1%	66.5%	61.0%
Vaccinated against rabies within last year	91.8%	84.2%	95.1%	87.2%	74.3%	86.3%	87.6%
Owned dog estimate	443,501	244,624	67,713	381,063	95,152	58,795	1,290,848
Human-to-dog ratio	12:1	4:1	19:1	3:1	14:1	10:1	8:1
Owned roaming dog estimate	208,174	162,460	43,219	279,139	34,659	29,674	757,325
Human-to-owned roaming dog ratio	26:1	6:1	30:1	4:1	39:1	20:1	14:1
Owned intact dog estimate	153,868	80,810	22,286	188,217	38,885	19,713	503,779

**Table 2 animals-15-01263-t002:** Number of CNVR rounds as predictors for the odds of an owned dog being allowed to roam for at least part of the day or night (binomial GLMM).

Roaming Behaviour (Allowed to Roam for at Least Part of the Day or Night)	Coefficient	Odds Ratio	95% Confidence Interval	*p*-Value
Sex (females vs. males)	0.088			0.63
Age in years	−0.061	0.94	0.89–1.00	0.04 *
1 round of CNVR vs. control (0 CNVR)	−0.630	0.53	0.25–1.11	0.09 .
2 rounds of CNVR vs. control (0 CNVR)	−0.721	0.49	0.23–1.02	0.06 .
3 rounds of CNVR vs. control (0 CNVR)	−1.303	0.27	0.12–0.60	0.001 **

Significance: ‘**’ <0.01, ‘*’ <0.05, and ‘.’ <0.1.

**Table 3 animals-15-01263-t003:** Number of CNVR rounds as predictors of the odds of an owned dog being sterilised (binomial GLMM).

Sterilisation of Owned Dogs	Coefficient	Odds Ratio	95% Confidence Interval	*p*-Value
Sex (females vs. males)	1.179	3.25	1.24–4.71	<0.001 ***
Age in years	0.042			0.14
Roaming	0.406	1.50	1.02–2.19	0.04 *
1 round of CNVR vs. control (0 CNVR)	1.289	3.62	1.91–6.90	<0.001 ***
2 rounds of CNVR vs. control (0 CNVR)	1.693	5.43	2.79–10.60	<0.001 ***
3 rounds of CNVR vs. control (0 CNVR)	1.757	5.80	2.72–12.31	<0.001 ***

Significance: ‘***’ <0.001 and ‘*’ <0.05.

## Data Availability

The data presented in this study are available on request from the corresponding author. The data are not publicly available due to the consent agreement with participants.

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
