# Peer review of "Population Demographics of Owned Dogs in Greater Bangkok and Implications for Free-Roaming Dog Population Management"

_animals, 2025, doi:10.3390/ani15091263_

Round 1

Reviewer 1 Report

Comments and Suggestions for Authors

This is a very useful additional paper to the literature on dog owning populations and their health control and welfare in developing countries.  Only the following issues are raised

p. 6 line 226 - can more information be given around how age was determined - if age is only self determined by the owner of the dog, how accurate are these data particularly if over half the dogs are from the street and therefore origin and age unknown

p 12 line 322 are there any data showing correlation of the age of the dog with if it is allowed to roam/confined as data are presented on relationship of vaccination and sterilisation with roaming/ confined status 

p 13 line 385-7 are these discussions on the age results accurate given self determination of age by owner and small numbers of owners who knew the age of the dogs - this should be caveated with the clarity of age determination from the questions

Author Response

Thank you very much for taking the time to review our manuscript. Please find the detailed responses below and the corresponding corrections in track changes in the re-submitted files.

Comment 1: This is a very useful additional paper to the literature on dog owning populations and their health control and welfare in developing countries.  Only the following issues are raised.

Response 1: We are grateful that you have seen value in our manuscript and welcome this first positive comment.

Comment 2: p. 6 line 226 - can more information be given around how age was determined - if age is only self determined by the owner of the dog, how accurate are these data particularly if over half the dogs are from the street and therefore origin and age unknown.

Response 2: We agree that in the case of adopting an adult dog, the age may not be known, however many adoptions from the street would have been of puppies, so age would have been known within a 1-2 month range. Owners were not encouraged to guess and if they did not know, this was recorded as ‘unknown’ for 31% of the dogs.

Revision 2: We have clarified the unknown category in the manuscript and stated the % of dogs for which age was known.

Comment 3: p 12 line 322 are there any data showing correlation of the age of the dog with if it is allowed to roam/confined as data are presented on relationship of vaccination and sterilisation with roaming/ confined status.

Response 3: In line 246 and in table 2 we include the relationship between age and roaming - “Increasing age was associated with lower odds of roaming, although the effect size per year of increasing age was small (OR 0.94, 95% CI 0.89-1.00, p < 0.05; Table 2).”.

Comment 4: p 13 line 385-7 are these discussions on the age results accurate given self determination of age by owner and small numbers of owners who knew the age of the dogs - this should be caveated with the clarity of age determination from the questions.

Response 4: Most of the owners did know the age of their dogs (69% of dogs were reported to be of known age and sex). We recognise that respondents may not know the age of a dog who was adopted as an adult as opposed to puppy.

Revision 4: We have included additional text in the discussion to explain that this group is excluded, as recommended by the reviewer.  

Reviewer 2 Report

Comments and Suggestions for Authors

the manuscript studies the demography od domestic dogs in the Greater Bangkok area, the issue of free-roaming and/or non-domiciled dogs, and how this affects management policy. this study provides important insights on the local dynamics of dog ownership and spatial use, as well as having important implications for public health and management policy. However, there is the need for corrections and completing part of the manuscript. I provide my comments below.

General comments:

there is a part of the methods - the GLMMs - that is not presented in the results. Please, revise and include the results of the model selection in the results of the manuscript.

Abstract:

there are keywords that are already on the title (dog, free-roaming dog), I would recommend substituting these keywords to increase the indeaxability of the manuscript.

Methods

lines 177, 179: you can remove the quotation marks from "fixed" and "random" effects.

Please, include a passage explaining what is a "soi", at it is a term frequently used in the questions present in the questionnaire, and it is a term specific for the Thay context.

Please, include a table, or present a list of the outcome and predictor variables included for the construction of the GLMMs, inform what distribution was used (i.e.: gaussian, poission, binomial...), and how were the variables codified (yes/no, categories, counts...).

Please, cite what R packages were used for the GLMMs and AIC stepwise selection.

Section 2.6: (ethical approval): does this type of study do not require ethical approval from the local government? if so, please state in the text.

Results:

line 212: have the differences between provinces been statistically tested to be considered significantly different?

Table 1: please check the formatting of the table, as there is a lot of variation on the allignment of the values in the cells

Figure 3: please, add the meaning of the abbreviated terms in the figure legend, to avoid ambiguity

Author Response

Thank you very much for taking the time to review our manuscript. Please find the detailed responses below and the corresponding revisions in track changes in the re-submitted files.

Comment 1: there is a part of the methods - the GLMMs - that is not presented in the results. Please, revise and include the results of the model selection in the results of the manuscript.

Response 1: The use of binomial GLMM models are explained in section 2.5 data analysis. However, we agree it should be clearer that the odds ratios reported in tables 2 and 3 in the results section are from the application of these binomial GLMM models. 

Revision 1: We have added “binomial” outcome in section 2.5 data analysis to make it clear that our outcomes were 0/1 – and clarified these 0/1 outcomes in the sentence that follows. We have added (“binomial GLMM”) in brackets to tables 2 and 3 headings where these results are reported to make it clear this was the statistical model used.

Comment 2: there are keywords that are already on the title (dog, free-roaming dog), I would recommend substituting these keywords to increase the indeaxability of the manuscript.

Response 2: We had not appreciated that indexing algorithms use both the title and keywords. And have asked the MDPI editors for guidance here, as this is not immediately obvious from the author guidelines.

Comment 3: lines 177, 179: you can remove the quotation marks from "fixed" and "random" effects.

Response 3: Agreed.

Revision 3: Deleted the quotation marks from the methods section, as suggested by the reviewer.

Comment 4: Please, include a passage explaining what is a "soi", at it is a term frequently used in the questions present in the questionnaire, and it is a term specific for the Thay context..

Response 4: Agreed.

Revision 4: Thank you for this suggestion, we have added a sentence to section 2.2 survey design that explains the term “soi”.

Comment 5: Please, include a table, or present a list of the outcome and predictor variables included for the construction of the GLMMs, inform what distribution was used (i.e.: gaussian, poission, binomial...), and how were the variables codified (yes/no, categories, counts...).

Response 5: Agree that this needs to be clearer.

Revision 5: We have clarified that a binomial distribution was used, and have provide a written explanation of the 0/1 outcomes in section 2.5 data analysis.

Comment 6: Please, cite what R packages were used for the GLMMs and AIC stepwise selection.

Response 6: Agreed.

Revision 6: We have added the package name to the methods section.

Comment 7: Section 2.6: (ethical approval): does this type of study do not require ethical approval from the local government? if so, please state in the text.

Response: There's was no clear way to get local ethical approval because none of the co-authors are affiliated with a Thai institution that has an ethical review board. The Dogs Trust ethical review board was developed to address this situation, providing oversight in terms of best practice research standards and ethics to organisations investing resources in objective data collection, analysis and interpretation outside of the usual academic/research environment where such ethical review boards are normally situated. This Dogs Trust ethical review board is experienced at applying global ethical standards of research (such as the Declaration of Helsinki) to a range of contexts within and outside of the UK. However, we do agree that a local review board would have been ideal and are now working to establish local links that can support us in this way in future – we thank you for inspiring us to work on this.

Comment 8: line 212: have the differences between provinces been statistically tested to be considered significantly different?

Response 8: Provincial differences were statistically tested and were significant for some, but not all of the many results presented in table 1 - this involved so many tests of statistical significance that reporting them all was cumbersome and hence we left them out of this table.

Comment 9: Table 1: please check the formatting of the table, as there is a lot of variation on the allignment of the values in the cells.

Response 9: Agreed. Thank you for spotting this.

Revision 9: We have amended the table.

Comment 10: Figure 3: please, add the meaning of the abbreviated terms in the figure legend, to avoid ambiguity.

Response 10: Agreed. Thank you for this suggestion.

Revision 10: We have added definitions as suggested.

Reviewer 3 Report

Comments and Suggestions for Authors

In the manuscript, the authors examined the Population demographics of owned dogs in Greater Bangkok, and implications for free-roaming dog population management.
However, the following comments can be made.
1. Abstract. More attention should be paid to the evaluation of the results obtained. Describe the results obtained. Add the purpose of the study.
2. Introduction. There is no purpose of the study.
3. Materials and methods. There is no information about the sample size. Also add information about how the survey was conducted, how the respondents were selected.
4. Conclusion. Evaluation of the obtained data in the conclusion is not enough. Supplement.

Author Response

Comment 1: Abstract. More attention should be paid to the evaluation of the results obtained. Describe the results obtained. Add the purpose of the study.

Response 1: Agreed.

Revision 1: We have added a short summary of the key results to the simple summary and amended the subsequent sentence to clarify the purpose of the study. In the abstract, we added text to the Abstract to be more explicit about the purpose of the study (Lines 23-26). With the 200-word limit for the Abstract, we unfortunately do not have space to describe the results beyond the “key findings” already referenced (Lines 33-37)

Comment 2: Introduction. There is no purpose of the study.

Response 2: We respectfully suggest that the purpose of the study is described in Lines 60-65 (in the revised version of the manuscript). A more in-depth discussion of the purpose of and justification for the study is in the second-to-last paragraph of the Introduction (Lines 95-112).

Comment 3: Materials and methods. There is no information about the sample size. Also add information about how the survey was conducted, how the respondents were selected.

Response 3: The CNVR project aims for 80% coverage, so we wanted a sufficient number of owned dogs in our sample to calculate with 95% confidence, and 5% error whether we had reached that target coverage in the owned dog population – this requires a sample of 246 owned dogs. We assumed 20% of the households would own an average of 1.5 dogs, hence we needed a sample of approximately 800 households from each of the 4 categories to reach this number of owned dogs.

Revision 3: We have added an explanation about the sample size to section 2.4. We have also added some addition detail on household selection to the same section.

Comment 4: Conclusion. Evaluation of the obtained data in the conclusion is not enough. Supplement.

Response 4: Agreed.

Revision 4: We have added a sentence noting the impact of CNVR on owned dog sterilisation and roaming as evaluation of CNVR impact was part of the purpose of this study.

Round 2

Reviewer 2 Report

Comments and Suggestions for Authors

The authors have produced a significant, detailed and well-signaled revision of the manuscript, based in the comments and suggestions presented in the first round of peer-review, resulting in a substantially improved manuscript. I consider that the current version is fit for publication and does not require further revisions.